# Components related to ethical decision making in medical science students: A structural equation model

Soraya Parvari[1], Hoorvash Farajidana[2,3]*, Zohreh Mahmoodi[4]*, Farima Mohamadi[5], Kourosh Kabir[6], Mehrad Boostanchi[7], Nami Mohammadian Khonsari[8]

1 Department of Anatomy, Faculty of Medicine, Alborz University of Medical Sciences, Karaj, Iran, 2 Department of Clinical Toxicology, Kosar Hospital Poison Center, Emergency Department, Alborz University of Medical Science, Karaj, Iran, 3 Department of Forensic Medicine and Medical Ethics, Alborz University of Medical Sciences, Karaj, Iran, 4 Social Determinants of Health Research Center, Alborz University of Medical Sciences, Karaj, Iran, 5 Social Determinants of Health Research Center, Shahid Beheshti University of Medical Sciences, Tehran, Iran, 6 Department of Community Medicine, School of Medicine, Alborz University of Medical Sciences, Karaj, Iran, 7 Student Research Committee, Faculty of Medicine, Alborz University of Medical Sciences, Karaj, Alborz, Iran, 8 Non-Communicable Diseases Research Center, Alborz University of Medical Sciences, Karaj, Iran

* f.drhoorvash@yahoo.com (HF); zohrehmahmoodi2011@gmail.com (ZM)

**Data Availability Statement:** All relevant data for this study are publicly available from the Zenodo repository (https://doi.org/10.5281/zenodo.10012885).

## Abstract

### Background

Evaluating moral principles in the Society of Medical Sciences and health care workers (HCWs) is imperative due to their direct contact with the community and the significant impact of their attitudes and decisions on people's lives. This study aimed to determine the components related to ethical decisions in medical sciences students.

### Methods

One thousand two hundred thirty-five eligible students in the Alborz University of Medical Sciences participated in this descriptive study. We gathered their socio-demographic information, assessed their moral reasoning, and used the ethical decisions questionnaire, Lutsen moral sensitivity questionnaire, and general health questionnaire (GHQ) for data gathering. The data were analyzed with SPSS software version 25 and LISREL version 8.8.

### Results

According to the path analysis test findings, ethical reasoning significantly correlated with ethical decision-making (B = 0.40). The number of clinical courses passed, moral sensation (moral sensitivity), and the total number of passed academic semesters had the greatest positive and negative association with ethical decision-making, respectively. (B = 0.54), (B = 0.524) and (B = -0.11).

**Funding:** he author(s) received no specific funding for this work.

**Competing interests:** The authors have declared that no competing interests exist.

## Conclusion

Based on the findings of the moral reasoning test, the moral sensation was associated with ethical decision-making, which indicates the necessity of attending to ethical aspects, promoting moral reasoning, sensitivity, and students' accuracy.

## Background

Ethics is a branch of philosophy and a vast idiom covering studies from moral nature to moral decision making [1]. The significant developments in science, medical, and environmental technologies in the present era have resulted in discussing and exchanging views on ethical issues and decision-making [2].

Medical Science professionals need ethical knowledge as a guideline. They are required to abide by ethics in any circumstance. In other words, ethics is one of the fundamental parts of Clinical Sciences [1].

Medical sciences students must adhere to ethics with respect to patients' values and needs [3]. ethical decision-making is applied to the totality of the decision-making process resulting from the detection of ethical issues through moral reasoning [4]. In moral dilemmas that put moral matters in conflict, ethical decision-making results from analyzing possible approaches and assessing possible acts to make a conscious decision to participate in an action [5]. Hence teaching moral reasoning alongside moral principles is vital [4]. Several factors can affect the ethical decision-making process (e.g., cultural differences, religious beliefs, experience, age, individual differences, commitment, mental health, and the individual's sensitivity) [6].

Furthermore, many researchers believe that ethical decision-making is not merely the result of moral reasoning, but mediating factors such as emotions, motivation, mental health, and social factors play a part as well [7]. Nonetheless, the relationship between the aforementioned variables and ethical decision-making is not constant; since many studies show conflicting results [8]. Hence, with regards to the position of students of Medical Sciences and their vital role in community health, we performed this structural equation model to determine the components affecting moral decision-making.

## Methods

This descriptive-analytical study was conducted on all available university students of Alborz University of Medical Sciences in Alborz province, Iran, in 2021.

### Sample size and participants

By extracting and assessing the Means and standard deviations of similar studies, a standard deviation of 10 for moral sensitivity and the maximum error of 2.5 in the mean, a sample size of 171 participants was needed furthermore with the presumption of 10% loss in the participants, 205 participants were needed in each group.

Considering the five groups of medical sciences students in the university (nursing and midwifery, operating room, and anesthesia, and medicine), a minimum of 1025 participants were needed for the study.

## Inclusion and exclusion criteria

All available medical Science students (nursing, midwifery, operating room, anesthesia, and medicine), with at least one year of clinical course experience, and a minimum of 6 hours per day hospital/clinical shifts who were willing to participate could enter the study. Incomplete completion of questionnaires, withdrawal from the study, transferring from the university during the study were our exclusion criteria.

## Data collection

In the present study, we used ethical decision-making, moral reasoning, and Lutsen moral sensitivity questionnaires, general health questionnaires (GHQ), and a socio-demographic data checklist.

**1. The ethical decision-making questionnaire.** This questionnaire contains 20 questions and determines the level of ethics in people; the research team confirmed its reliability and validity. The present study determined the questionnaire's Cronbach's Alpha as 0.90 [9].

**2. The Lutsen moral sensitivity questionnaire.** This questionnaire consists of 25 questions and six retail scales, including the amount of respect for the participant's independence, the amount of knowledge of how to communicate with the participant, the amount of professional knowledge, experience in ethical dilemmas, application of the ethical concepts in moral decision making, honesty, and benevolence.

The scoring is based on a five-point Likert scale (from 0 meaning completely disagree to 4 meaning completely agree). The total score is calculated by the sum of the scores in each question ranging from 0 to 100.

The validity and reliability of the Persian version of the questionnaire in the Iranian was investigated by Hassanpour and colleagues with estimated reliability of 0.81 [10].

**3. The moral reasoning questionnaire.** This questionnaire is designed based on Kolberg's theory of evolution (including three short stories of everyday moral dilemmas in the society. The scoring is based on a five-point Likert scale ranging from 0 meaning very low to 4 meaning very high. The total score is calculated by the sum of the scores of stories. The validity and reliability of the Persian version of the questionnaire in the Iranian population was assessed by Marzban Moradi et al. with an estimated Cronbach's Alpha of 0.77 [11].

**4. The general health questionnaire.** GHQ consists of 28 questions and was first presented by Goldberg and Hiller (1979) [12], this questionnaire has four sub-scales, and each scale has seven questions. The aforementioned scales are the Physical symptoms scale, anxiety symptoms scale, sleep disorder, social function scale, and depression symptoms scale. From the 28 questions, 1 to 7 correspond to the physical symptoms scale, 8 to 14 to anxiety and sleep disorder, 15 to 21 to social function evaluation, and 22 to 28 cases to depression. (Scoring is based on a Likert scoring scale ranging from 0 to 3). The cut-off point of the sum of the total score in this questionnaire is 23 suggesting the presence of possible mental issues in scores above 23.

The validity and reliability of the Persian version of the questionnaire in the Iranian population were assessed with an estimated Cronbach's Alpha coefficient of 0.83 [13].

## Socio-demographic checklist

This checklist consisted of demographic and socioeconomic status (SES) questions.

Eslami et al. estimated that the checklist's Cronbach's Alpha to be 0.83 [14].

## Study design

After obtaining the necessary permits and ethical approval from the Alborz University of medical sciences ethics committee, due to the conditions that resulted from the COVID-19 pandemic and the lockdowns, the consent form regarding participation in the study and the questionnaires were sent through the internet with the networking aid of the student affairs, Student Research Committee and student groups in WhatsApp and Telegram. The researchers' contact information was provided to the students to respond to possible ambiguities.

The students were ensured that all their information would be kept confidential and participation within the study was completely voluntary.

## Ethics approval and consent to participate

The Ethic Committee of Alborz University of Medical Sciences approved this study written informed consent was obtained from all participants via the internet. Furthermore, all methods were performed in accordance with the relevant guidelines and regulations.

## Statistical analysis

In this study, the fitness of a conceptual model of the simultaneous relationship between ethical decision-making components was investigated in medical science students. We assessed the normal distribution of continuous variables using the Kolmogorov-Smirnov test.

Variables were analyzed using SPSS 25 [9], PLS3 [10], and Lisrel 8.8 [13] software. Correlation between continuous variables was assessed using the Pearson correlation coefficient. The results of the structural equation model are reported as beta (β) coefficient and T-value. T-values > 1.96 were considered statistically significant.

# Results

In the present study, 1235 students comprising of 237 medicine, 276 nursing, 261 midwifery, 229 anesthesiology and 232 operation room students were evaluated. Questionnaires were sent to all the 1235 students and all were gathered due to our consistency in data collection. The average age of the participants was 24.6 ± 3.8 ranging from 20 to 40 years. The average scores of the ethical decision-making, moral reasoning, mental health, and moral sensitivity were 63.3, 118.6, and 52.6, respectively (Table 1).

**Table 1. Demographic information distribution in clinical students of Alborz University of Medical Sciences.**

| Variable | | Number | Percent |
|---|---|---|---|
| **Age** | **18–23** | 595 | 49 |
| | **24–29** | 410 | 33.7 |
| | **30–35** | 204 | 16.8 |
| | **36 And Above** | 6 | 0.5 |
| **Sex** | **Male** | 613 | 49.7 |
| | **Female** | 621 | 50.3 |
| **Marital Status** | **Single** | 1001 | 81.1 |
| | **Married** | 234 | 18.9 |
| **Major** | **Medicine** | 237 | 19.2 |
| | **Nursing** | 276 | 22.3 |
| | **Midwifery** | 261 | 21.1 |
| | **Anesthesiology** | 229 | 18.8 |
| | **Operating Room** | 232 | 18.5 |

**Table 2. Pearson's correlation between age, education, number of passed academic and clinical semesters, moral reasoning, sensation, mental health, SES and ethical decision making.**

| | Age | Education | Total Passed Semesters | Number Of Passed Clinical Semesters | Moral Sensation | Moral Argument | Moral Decision Making | Mental Health |
|---|---|---|---|---|---|---|---|---|
| **Age** | 1 | | | | | | | |
| **Education** | 0.693** | 1 | | | | | | |
| **Total Passed Semesters** | 0.201** | 0.492** | 1 | | | | | |
| **Number Of Passed Clinical Semesters** | 0.312** | 0.605** | 0.933** | 1 | | | | |
| **Moral Sensation** | 0.097** | 0.118** | 0.029 | 0.072* | 1 | | | |
| **Moral reasoning** | 0.337** | 0.358** | 0.152** | 0.233** | 0.319** | 1 | | |
| **Ethical Decision Making** | 0.196** | 0.208** | 0.098** | 0.159** | 0.527** | 0.503** | 1 | |
| **Mental Health** | 0.477** | 0.539** | 0.263** | 0.372** | 0.181** | 0.322** | 0.278** | 1 |
| **SES** | -0.104** | -0.136** | -0.120** | -0.125** | -0.085** | -0.161** | -0.121** | -0.130** |

*P-value < 0.05

**P-value<0.01

SES: socioeconomic status

The Pearson correlation test results showed that moral sensitivity, moral reasoning, mental health, and SES were significantly correlated with moral decision-making. In between the variables, moral sensitivity had the highest positive correlation (r = 0.527), and SES had the highest negative correlation (r = -0.121) with ethical decision-making (Table 2).

The path analysis test results showed that all variables were significantly correlated with ethical decision-making in one or both paths. Moral reasoning was the only variable with a single, significant positive path (B = 0.40). Variables such as age, mental health, number of passed clinical semesters, the number of passed academic semesters, and moral sensitivity were significantly correlated with ethical decision-making both direct and indirectly.

The number of clinical courses passed, moral sensitivity, and the total number of passed academic semesters had the greatest positive and negative association with ethical decision-making. (B = 0.54), (B = 0.524) and (B = -0.11) (Table 3). Furthermore, the path diagram for the association of socio-demographic parameters, moral sensation and moral reasoning with ethical decision making in the students is illustrated in Fig 1.

The results of the fitness model indices indicate the desirability, high proportionality of the model, and the logic of the adjusted relations of the variables based on the conceptual model.

**Table 3. Direct and indirect effects of socio-demographic parameters, moral sensitivity and moral reasoning with ethical decision making in the students.**

| Variables | Direct Effect | Indirect Effect | Total Effect | $R^2$ |
|---|---|---|---|---|
| **Age** | 0.13 | 0.2 | 0.33 | 0.8 |
| **Education** | 0.03 | 0.03 | 0.06 | |
| **Total Passed Semesters** | 0.21 | -0.324 | -0.114 | |
| **Passed Clinical Semesters** | 0.22 | 0.434 | 0.654 | |
| **SES** | 0.05 | -0.087 | -0.037 | |
| **Moral Sensitivity** | 0.42 | 0.104 | 0.524 | |
| **Moral Reasoning** | 0.4 | - | 0.4 | |
| **Mental Health** | 0.21 | 0.127 | 0.337 | |

SES: socioeconomic status

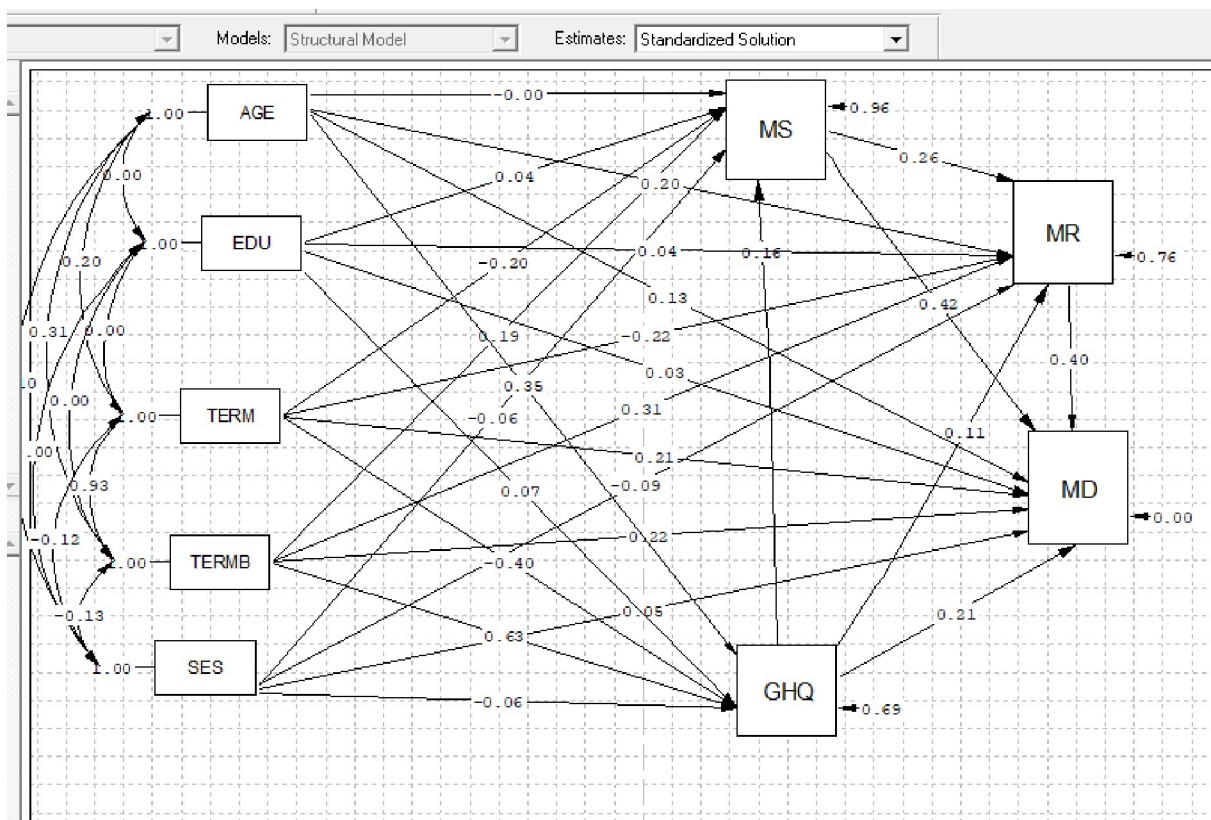

MR= moral reasoning,  MD= moral decision ,  MS= moral sensitive, TERM= total number of passed semesters,  TERMB=  number of passed semesters in hospital,      EDU=education,   GHQ=general Health Questionnaire.

**Fig 1. Path diagram for the association of socio-demographic parameters, moral sensation and moral reasoning with ethical decision making in the students.** MR = moral reasoning, MD = moral decision, MS = moral sensitive, TERM = total number of passed semesters, TERMB = number of passed semesters in hospital, EDU = education, GHQ = general Health Questionnaire.

Accordingly, the fitted model has no significant difference from the conceptual model (Table 4).

## Discussion

The findings of this study showed that ethical decisions are associated with several components.

**Table 4. The fitness model.**

| Model | X² | df | CFI | GFI | NFI | RMSEA |
|---|---|---|---|---|---|---|
| 8.56 | 4 | 2.14 | 1 | 0.91 | 1 | 0 |

Df: degree of freedom, CFI: Comparative Fit Index GFI: Goodness of Fit, NFI: Normed-Fit Index, RMSEA: Root Mean Square Error of Approximation.

Based on the study results, total passed academic semesters can directly decrease ethical decision-making; however, the number of passed clinical semesters increases the level of ethical decision-making. Based on these results, the presence within clinical courses is more important than the total passed academic semesters for the sake of ethical decision-making. Peirce et al. believe that the nature of reasoning is to consider what we have already learned and know and what we do not know. This subject is a reality and is not a reflection of an ideal subject based on results, and this is only possible when we are put in an actual situation and not just have an idea of the situation [15]. Clinical students face situations in which every decision that they make can affect the life of their patient; it is in these circumstances that they can learn moral reasoning and understand the fundamental aspects of ethics [16, 17].

Qin Chen et al. also reported that individual and social characteristics are the main difference in ethical decision-making, and the more thorough the methods of clinical education are, the students' ability in ethical decision-making increases [18].

In this study, moral sensitivity had a positive direct and indirect association with ethical decision-making. Hence moral sensitivity had a direct relationship with moral reasoning. Moral sensitivity is considered as the combination of awareness of the moral dimension (e.g., peace and prosperity, responsibility, attending to ethical issues) that reflects the concerns of a person regarding their actions towards others and helps them to find the correct path [19].

Lützén et al. believe that students' moral sensitivity results in timely recognition of the patients' problems, and thus the appropriate action in a timely manner towards treatment can be taken [20]. Furthermore, research shows that professional nurses with a higher level of moral sensitivity are better caregivers and are more respectful with their colleagues [21].

Regarding the positive correlation of moral reasoning with the level of ethical decision-making found in this study, Moral reasoning can be a response to how to behave in a real or hypothetical moral dilemma. For example, in a situation merged with the rules and principles of ethics in which choosing an answer is necessary, a judgment or assessment regarding the ethical acceptability of actions or the moral qualities of others. (e.g., the judgment of individuals, groups, or institutions). [22] in general, moral reasoning is the ability to select a solution among several solutions in a moral dilemma [23]. A study by Joolaee et al. revealed that during caregiving, nurses consider ethical dilemmas; however, the lack of proper support and ethical instructions results in inadequacies in moral reasoning and ethical decision making [24]. Also, Sari et al. found that students with a lower level of moral reasoning have difficulties in ethical decision-making [25]. Furthermore, Ashoori et al. found a significant relationship between ethics, moral reasoning, and ethical decision-making in nurses as well [26].

This study found that mental health has a positive relationship with moral decision-making, whether directly or indirectly. A decisive ethical decision requires a proper level of mental health [27]. Ebrahimi et al. evaluated the psychological reactions of nurses regarding ethical decision making; they found that ethical decisions from which the patients benefit, results in positive psychological reactions in nurses (e.g., satisfaction sensation, increased motivation, and sense of competence); nonetheless, inadequacies and unfavorable results and negative outcomes result in adverse psychological reactions (e.g., stress, anxiety and depression, dissatisfaction, decreased motivation and demerit) [28].

## Conclusion

Based on the findings of the moral reasoning test, moral sensation was associated with ethical decision-making, which indicates the necessity of attending to ethical aspects, promoting moral reasoning, sensitivity, and students' accuracy.

## Author Contributions

**Conceptualization:** Soraya Parvari, Hoorvash Farajidana, Kourosh Kabir.

**Data curation:** Zohreh Mahmoodi.

**Formal analysis:** Zohreh Mahmoodi, Farima Mohamadi.

**Investigation:** Soraya Parvari, Hoorvash Farajidana, Mehrad Boostanchi.

**Methodology:** Zohreh Mahmoodi, Kourosh Kabir.

**Resources:** Soraya Parvari.

**Writing – original draft:** Nami Mohammadian Khonsari.

**Writing – review & editing:** Nami Mohammadian Khonsari.

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
