## [Decision Letter · Decision Letter 0]

13 Sep 2023

PONE-D-23-05611Components related to ethical decision making in medical science students: A Structural Equation Model.PLOS ONE

Dear Dr. Mahmoodi,

Thank you for submitting your manuscript to PLOS ONE. After careful consideration, we feel that it has merit but does not fully meet PLOS ONE’s publication criteria as it currently stands. Therefore, we invite you to submit a revised version of the manuscript that addresses the points raised during the review process.

We look forward to receiving your revised manuscript.

Kind regards,

Fatma Ay, Ph.D

Academic Editor

PLOS ONE

Journal Requirements:

Reviewers' comments:

Reviewer's Responses to Questions

**Comments to the Author**

1. Is the manuscript technically sound, and do the data support the conclusions?

Reviewer #1: Yes

Reviewer #2: Yes

2. Has the statistical analysis been performed appropriately and rigorously? 

Reviewer #1: I Don't Know

Reviewer #2: No

3. Have the authors made all data underlying the findings in their manuscript fully available?

Reviewer #1: Yes

Reviewer #2: Yes

4. Is the manuscript presented in an intelligible fashion and written in standard English?

Reviewer #1: Yes

Reviewer #2: Yes

5. Review Comments to the Author

Reviewer #1: This manuscript could make a valuable contribution to literature on ethics.

In result, it would be better if minimum and maximum age is also included along with average age of the participants

In conclusion, write comma (,) after the statement 'Based on the findings of the moral reasoning test'

Reviewer #2: In the methods sections the words patients and participants are used interchangeably. The rest of the text suggests that the appropriate term to use is participants. Please correct accordingly.

The Cronbach alpha of the Ethical decision Making questionnaire was reported as 90. Probably this should read 0.9 since it ranges from 0 to 1.

There were 5 groups of students that participated in the study namely (nursing and

midwifery, operating room, and anesthesia, and medicine). The number of participants in each sub group should be described. Eg How many nursing students participated in the study? Additionally, the sampling method used should be described elaborately. Were questionnaires sent to all students that are enrolled? If so what was the return rate? It is possible that the course that the student is enrolled in has an impact on the scores. This should be evaluated. A simple linear regression model can suffice. Different student groups have different exposure to the clinical setting. This exposure varies both in terms of actual time spend with the patients as well as the nature of the contact with the patients. It is therefore important to investigate whether the relationships studied hold for the different student subgroups or only some of them.

6. PLOS authors have the option to publish the peer review history of their article (what does this mean?). If published, this will include your full peer review and any attached files.

Reviewer #1: No

Reviewer #2: No

---

## [Author Response · Author response to Decision Letter 0]

17 Oct 2023

Reviewer #1: This manuscript could make a valuable contribution to literature on ethics.

In result, it would be better if minimum and maximum age is also included along with average age of the participants

answer: thank you for the point, added

In conclusion, write comma (,) after the statement 'Based on the findings of the moral reasoning test'

answer: thank you for the point corrected

Reviewer #2: In the methods sections the words patients and participants are used interchangeably. The rest of the text suggests that the appropriate term to use is participants. Please correct accordingly.

Answer: thank you for the comment, patient was substituted with participant in relevant sections within the methods 

The Cronbach alpha of the Ethical decision Making questionnaire was reported as 90. Probably this should read 0.9 since it ranges from 0 to 1.

Answer: thank you for this very important comment. This was a typo error and was corrected. 

There were 5 groups of students that participated in the study namely (nursing and

midwifery, operating room, and anesthesia, and medicine). The number of participants in each sub group should be described. Eg How many nursing students participated in the study?

answer: thank you for the point, added

 Additionally, the sampling method used should be described elaborately. Were questionnaires sent to all students that are enrolled? If so what was the return rate? 

answer: thank you for the important point. We clarified this in the results. Due to the fact that all students who were willing to participate were already available and our consistency in data collection we gathered all the questionnaires sent to the 1235 students. 

It is possible that the course that the student is enrolled in has an impact on the scores. This should be evaluated. A simple linear regression model can suffice.

Answer: Thank you for the important point.

 Path analysis is a statistical technique that discern and assess the effects of a set of

variables acting on a specified outcome via multiple causal pathways. This method

allows users to investigate patterns of effect within a system of variables. It is one of

several types of the general linear model that examine the impact of a set of

predictor variables on multiple dependent variables. (1)

In MUNRO’S Statistical Methods for Health Care Research book (2013) explained

that: Path analysis is a type of multiple regression statistical analysis used to examine

causal models by examining the relationships between a dependent variable and two

or more independent variables..Multiple regression analysis is also used to assess

whether confounding exists. Since multiple linear regression analysis allows us to

estimate the association between a given independent variable and the outcome

holding all other variables constant, it provides a way of adjusting for (or accounting

for) potentially confounding variables that have been included in the model.(2)

1- Jupp V. The Sage dictionary of social research methods: Sage; 2006.

2- Plichta SB, Kelvin EA, Munro BH. Munro's statistical methods for health care research: Wolters

Kluwer Health/Lippincott Williams & Wilkins; 2013

thus the results of the path analysis can be considered as adjusted linear regression findings 

 Different student groups have different exposure to the clinical setting. This exposure varies both in terms of actual time spend with the patients as well as the nature of the contact with the patients. It is therefore important to investigate whether the relationships studied hold for the different student subgroups or only some of them.

Answer: thank you for the important point, we completely agree with the comment, the nature of exposure and time spent in the clinics may differ. Nonetheless the questions within the questionnaire is not based on major nor time spent within the clinics, they are questions regarding cases and ethical dilemmas that assess the answerer’s decision making in that situation. However to minimize the time difference and clinical experience of students, only student’s with at least one year of clinical experience who spent 6 hours daily in the clinics or hospitals were included in the study (this was added to the methods). 

Furthermore, although it is possible to present the findings for each group separately, it will significantly lengthen the results. However, with regard to the answer above if the reviewer feels that it is still necessary to investigate the relationships based on students’ majors we will gladly add the relevant analyses and findings.

---

## [Decision Letter · Decision Letter 1]

8 Jan 2024

Components related to ethical decision making in medical science students: A Structural Equation Model.

PONE-D-23-05611R1

Dear Dr. Mahmoodi,

We’re pleased to inform you that your manuscript has been judged scientifically suitable for publication and will be formally accepted for publication once it meets all outstanding technical requirements.

Kind regards,

Fatma Ay, Ph.D

Academic Editor

PLOS ONE
---

## [Editor Report · Acceptance letter]

24 Jan 2024

PONE-D-23-05611R1 

PLOS ONE

Dear Dr. Mahmoodi, 

I'm pleased to inform you that your manuscript has been deemed suitable for publication in PLOS ONE. Congratulations! Your manuscript is now being handed over to our production team.

Kind regards, 

on behalf of

Dr. Fatma Ay 

Academic Editor

PLOS ONE